# Association between virtual visits and health outcomes of people living with HIV: A cross-sectional study

Nadia Rehman[1]*, Lawrence Mbuagbaw[1,2,3,4,5,6], Dominik Mertz[1,7,8☯], Giulia M. Muraca[1,9☯], Aaron Jones[1,10,11], on behalf of the Ontario HIV Treatment Network Cohort Study[¶]

1 Department of Health Research Methods, Evidence, and Impact, McMaster University, Hamilton, Ontario, Canada, 2 Department of Anesthesia, McMaster University, Hamilton, Ontario, Canada, 3 Centre for Development of Best Practices in Health (CDBPH), Yaoundé Central Hospital, Yaoundé, Cameroon, 4 Department of Pediatrics, McMaster University, Hamilton, Ontario, Canada, 5 Biostatistics Unit, Father Sean O'Sullivan Research. Centre, St Joseph's Healthcare, Hamilton, Ontario, Canada, 6 Division of Epidemiology and Biostatistics, Department of Global Health, Stellenbosch University, Cape Town, South Africa, 7 Faculty of Health Sciences, McMaster University, Hamilton, Ontario, Canada, 8 Department of Medicine, McMaster University, Hamilton, Ontario, Canada, 9 Department of Obstetrics and Gynecology, McMaster University, Hamilton, Ontario, Canada, 10 Institute for Clinical Evaluative Sciences (IC/ES), McMaster University, Hamilton, Ontario, Canada, 11 Institute for Clinical Evaluative Sciences; IC/ES, University of Toronto, Toronto, Ontario, Canada,

¶ Membership of the Ontario HIV Treatment Network (OHTN) is provided in the Acknowledgments.
☯ These authors contributed equally to this work.
* rehmann@mcmaster.ca

## Abstract

### Background

Virtual care has been integrated as a modality of care in Ontario, yet its effectiveness for people living with HIV remains largely unexplored.

### Objectives

We aimed to determine the association of visit modality (virtual, in-person, or both) on adherence to antiretroviral therapy (ART), viral load, and quality of life (QoL) in people living with HIV in Ontario, Canada.

### Methods

We conducted a cross-sectional study using data from the 2022 Ontario HIV Treatment Network Cohort Study (OCS), collected during the COVID-19 pandemic when virtual visits were first introduced. Participants were grouped into three categories based on the mode of care: virtual, in-person, or a combination of both. Data were collected through self-reported questionnaires and medical records, with viral load data linked to Public Health Ontario Laboratories (PHOL). Logistic regression was used to examine the outcomes of optimal ART adherence and viral load suppression, and linear regression was used for quality of life (mental and physical) outcomes.

**Data availability statement:** The data that support the findings of this study are not publicly available to protect the privacy of the participants. However, all aggregated data from the OHTN Cohort Study (OCS) can be made available to researchers upon reasonable request and access to line-level data can be obtained through a request to the OCS Governance Committee (https://ohtncohort-study.ca/research/). Requests to access data can be made by emailing the OCS coordinator via email (ocs@ohtn.on.ca).

**Funding:** The author(s) received no specific funding for this work.

**Competing interests:** The authors have declared that no competing interests exist.

## Results

In 2022, 1930 participants accessed HIV care in the OCS. Among them, 19.0% received virtual care, 45.6% received in-person care, and 34.3% received care through virtual and in-person modalities. The median age of the participants was 55 years (IQR: 45–62). In the multivariable logistic regression model, virtual care was associated with an increased likelihood of optimal adherence to antiretroviral therapy (Adjusted Odds Ratio (AOR) 1.30, 95% confidence interval (CI): 1.00, 1.70) and an increased likelihood of achieving viral load suppression (AOR 1.67, 95% CI:1.03, 2.63). Moreover, combined virtual and in-person care is associated with an improved mental quality of life compared to in-person care (Adjusted Mean difference (MD) -0.960, 95% CI: 0.05, 1.87).

## Conclusion

This study suggests virtual care is positively associated with adherence to antiretroviral therapy (ART) and viral suppression within this context. However, future research is necessary to establish causality and to assess the long-term effects of virtual care.

## Introduction

HIV is a chronic health challenge, with 22,461 people living with HIV in Ontario, Canada, as of 2020 [1]. Socio-economic disparities and structural barriers complicate patient-provider relationships, hindering continuity of care, retention in healthcare services, and adherence to antiretroviral therapy (ART) [2]. This results in detectable viral loads, increased opportunistic infections, and higher morbidity and mortality rates [3–6]. Despite universal public healthcare coverage, retaining individuals in care remains challenging, highlighting the need for a fully accessible, patient-centred system [7–10].

To address healthcare access issues, Ontario adopted virtual care in 2021 during the SARS-CoV-2 pandemic, expanding it in 2022 to complement traditional care [11]. Virtual care is a health care model in which all clinical interactions between the practitioner and the patient are delivered using electronic mediums, such as video conferencing or audio digital tools, such as telephone [12]. Since 2022, Ontario has taken concrete measures to integrate virtual visits, including strengthening data security and privacy, addressing social and ethical concerns, establishing virtual healthcare regulations, and introducing fee codes for physician billing.[10,11].

Virtual visits can enhance HIV care by improving convenience, accessibility, and affordability, potentially increasing patient retention [13,14]. However, there are limitations with the use of virtual care, including the inability to conduct physical assessments, lack of necessary equipment, and reduced in-person interactions, which may lead to mistrust, multiple appointments, and misdiagnosis [13]. Technology can also be a barrier for older adults, individuals with lower literacy, and those from disadvantaged socio-economic backgrounds [14]. Physician preferences and discrepancies

in Ontario's fee codes between comprehensive and limited care and telephone versus video visits further influence virtual care use [10,15]. Evidence on the effect of virtual care on health outcomes for people living with HIV (PLHIV) is limited. While some studies focus on retention improvement rather than health outcomes, findings are mixed—some suggest virtual care supports retention, while others indicate higher loss to follow-up[12,16–18]. As virtual care evolves, addressing disparities, preventing overuse, and considering financial and ethical implications remain crucial [10,12,16,19,20].

Although virtual care is now standard practice in Ontario [10], tits effectiveness remains uncertain. This study, conducted in collaboration with the Ontario HIV Treatment Network (OHTN) [21], utilizing the data from the Ontario HIV Treatment Network Cohort Study (OCS), North America's largest community-governed HIV cohort [22]. The primary objective of the study was to assess whether there were differences in adherence to antiretroviral therapy (ART), quality of life (QoL), and viral load among people living with HIV in Ontario, Canada, based on whether they used virtual or in-person appointments with an HIV care physician. The study focused on how different modalities relate to the health outcomes of PLHIV. The secondary objective of the study was to evaluate the differences in health outcomes (adherence to ART, QoL, and viral load) among people living with HIV from various socio-demographic and health-related factors in Ontario, as virtual care may affect certain groups differently.

## Methods

### Study design

We conducted a cross-sectional study using data from participants in the OCS in 2022.

### Setting

The OCS is a multi-site clinical cohort of people receiving HIV care in Ontario, Canada's most populous province (population: 13.6 million). Recruitment occurs at ten participating sites, including outpatient clinics in hospitals and community-based practices. The cohort has been described elsewhere [23].

### Stakeholder engagement

To achieve our study objectives, we established a community advisory board (CAB) in collaboration with Realize, a Canadian charitable organization working with people living with HIV and related organizations [24]. The CAB comprises representatives from various key populations of people living with HIV, enhancing the external validity of our project, bolstering individual and community capacity, and ensuring the effective implementation of our research findings [25,26]. We convened a meeting with the CAB to seek their insights on the relevance of the research question, the socio-demographic factors involved, and the correlation with the health conditions of people living with HIV. The CAB also actively interpreted the findings and collaborated on a dissemination plan for our research outcomes.

### Data sources

Clinical data are collected during routine follow-up visits, sourced from clinic records via manual chart abstractions or computerized medical record systems, and record linkage with Public Health Ontario Laboratories (PHOL), the sole provider of such testing provincially. Additionally, annual interviews are conducted using standardized questionnaires to gather socio-demographic and psycho-social-behavioural information.

### Ethics

All participants provide written informed consent, and the cohort design and consent forms are approved by the University of Toronto Research Ethics Boards (REBs) and REBs at each participating site [23]. Data was accessed on November 23, 2023. And the data was in de-identified form.

## Population

Eligible participants for our study are those aged 16 years or older who visited their HIV physician using any of the three modalities of care (virtual, in-person visits, or both virtual and in-person care) in 2022 and completed the OCS questionnaire. Participants with incomplete information on the type of care received were excluded.

## Measures/outcomes

**Primary outcome.** Adherence to ART is measured by self-report (never skipped, within the past week, 1–2 weeks, 2–4 weeks ago, 1–3 months ago, more than three months ago) in the standardized OCS questionnaire. This questionnaire is part of the Adult AIDS Clinical Trials Group (ACTG) adherence assessment and has been validated in previous studies [27–29].Self-reported nonadherence has shown high specificity, indicating its clinical significance and the need for further discussions between providers and patients [27]. For this study, we dichotomized adherence into optimal adherence (≥95%) and suboptimal adherence (<95%). We considered optimal adherence as participants who reported never missing a dose or those who missed a dose more than three months ago. In contrast, suboptimal adherence included those who missed doses within the past week, 1–4 weeks ago, 1–3 months, or responded with "don't know." These classifications were established in consultation with HIV specialists from a dedicated HIV care facility.

**Secondary outcomes.** Viral load suppression was defined as ≤ 40 copies/mL, indicating undetectable viral load (viral suppression) [30]. QoL was assessed using the Short Form 12-item Health Survey (version 2), which includes the Mental Component Summary Score (MCS) and the Physical Component Summary Score (PCS). Both scores are reported separately, with higher values indicating better QoL in their respective domains [31].

## Variables

**Primary exposure.** We categorized the HIV care patients received into three mutually exclusive categories: i). In-person care in the clinic ii). Virtual care either by telephone or video call, iii). Received both in-person and virtual forms of care. We defined virtual care as visits with the HIV care physician by telephone or video call, as defined in the OCS Questionnaire 2022.

**Demographic and clinical variables.** Baseline demographic and clinical data were extracted from the 2022 OCS questionnaire. Due to multiple categories in the OCS data, some of which were not information-rich, we consolidated them into meaningful categories for our study. Data missing values are reported separately in Table 1, which details the baseline characteristics.

For demographic variables, we combined sex (female versus male) and sexual orientation (men who have sex with men (MSM) vs non-MSM) to derive a single variable named sex, compromising three categories: females, male MSM, and male non-MSM and relationship status categorized as stable (married, living common-law, living in a committed relationship) vs unstable relationship (widowed, separated/divorced, single).

Clinical variables were extracted on the following health conditions: alcohol use disorder syndrome use was defined as on the Alcohol Use Disorder Identification Test (AUDIT-10) with harmful alcohol measured from 10- items (a score of ≥8 regardless of gender/sex) [32], depression measured by the Patient Health Questionnaire (PHQ) scale with nine items [33], and diagnosis of mental health comorbidities was based on self-report in the OCS question.

We used a revised 10-item HIV-related stigma scale categorized into four major components of HIV-related stigma: personalized stigma, worries about disclosure of status, negative self-image, and sensitivity to public reactions about HIV status. Individuals who responded with "agree' and "strongly agree" were identified as experiencing stigma in at least one of the four components [34].

**Table 1. Comparison of baseline characteristics between participants who accessed HIV care through virtual, in-person or both virtual and in-person care in Ontario, Canada, in 2022.**

| Characteristics | In-person care n (%) | Virtual care n (%) | Virtual and in-person care n (%) | Total N (%) | p-value |
|---|---|---|---|---|---|
| **No. of participants** | 900 | 367 | 663 | 1930 | |
| *Median age [Q1-Q3]* | 55 [44-63] | 56 [46–62] | 55 [45-62] | 55 [45-62] | 0.772 |
| *Sex* <0.001 | | | | | |
| Female | 233 (25.9) | 60 (16.3) | 132 (20.2) | 425 | |
| Male MSM [a] | 459 (50.8) | 253 (70.0) | 419 (63.4) | 1131 | |
| Male non-MSM | 203 (22.6) | 49 (13.4) | 110 (16.2) | 362 | |
| Missing | 5 (0.5) | 5 (1.1) | 2 (0.3) | 12 | |
| *Race* <0.001 | | | | | |
| White | 478 (53.1) | 252 (68.7) | 442 (66.6) | 1172 | |
| Black | 263 (29.3) | 55 (14.9) | 130 (19.6) | 448 | |
| Other | 159 (17.6) | 60 (16.3) | 91 (13.7) | 310 | |
| *Region* <0.001 | | | | | |
| Eastern Ontario | 127 (14.1) | 24 (6.5) | 22 (3.3) | 173 | |
| Northern Ontario | 18 (2.1) | 5 (1.4) | 2 (0.3) | 25 | |
| Southwestern Ontario | 99 (11.2) | 62 (16.9) | 174 (26.2) | 335 | |
| Toronto | 656 (72.8) | 276 (75.2) | 465 (70.1) | 1397 | |
| *Education* 0.246 | | | | | |
| Elementary | 25 (2.7) | 10 (2.7) | 14 (2.1) | 49 | |
| High school | 239 (26.5) | 126 (34.3) | 224 (33.7) | 589 | |
| College | 210 (23.3) | 64 (17.4) | 139 (20.9) | 413 | |
| Higher education | 363 (40.3) | 162 (44.1) | 284 (42.8) | 809 | |
| Missing | 63 (7.4) | 5 (1.4) | 2 (0.3) | 70 | |
| *Employment status* 0.285 | | | | | |
| Employed | 433 (48.1) | 182 (49.6) | 327 (49.3) | 942 | |
| Unemployed | 463 (51.4) | 182 (49.6) | 336 (50.7) | 981 | |
| Missing | 4 (0.4) | 3 (0.8) | 0 (0.0) | 7 | |
| *Gross annual income* 0.287 | | | | | |
| <50,000 | 374 (41.5) | 144 (39.2) | 280 (42.2) | 798 | |
| < 50,000-70,000 | 103(11.4) | 45 (12.2) | 75 (11.3) | 223 | |
| < 70,000-80,000 | 121 (13.4) | 41 (11.1) | 76 (11.4) | 238 | |
| >100,000 | 173 (19.2) | 94 (25.6) | 150 (22.6) | 417 | |
| Missing | 129 (14.3) | 43 (11.7) | 82 (12.3) | 254 | |
| *Relationship* 0.170 | | | | | |
| Stable | 376 (41.6) | 154 (42.0) | 246 (37.1) | 776 | |
| Unstable | 521 (57.8) | 212 (57.8) | 417 (62.9) | 1150 | |
| Missing | 3 (0.3) | 1 (0.3) | 0 (0.0) | 4 | |
| *Adherence to ART* 0.014 | | | | | |
| < 95% | 295 (32.7) | 137 (37.3) | 247 (37.3) | 679 | |
| ≥ 95% | 600 (66.6) | 225 (61.3) | 412 (62.1) | 1237 | |
| Missing | 5 (0.5) | 5 (1.3) | 4 (0.6) | 14 | |
| *Mental health conditions* 0.190 | | | | | |
| Depression | 197 (21.8) | 44 (12.0) | 167 (25.1) | 408 | 0.123 |
| *Alcohol use-disorder syndrome d* 0.248 | | | | | |
| **No** | 67 (7.4) | 34 (9.2) | 65 (9.8) | 166 | |
| **Yes** | 824 (91.6) | 331 (90.1) | 598 (90.1) | 1753 | |

*(Continued)*

**Table 1.** (Continued)

| Characteristics | In-person care n (%) | Virtual care n (%) | Virtual and in-person care n (%) | Total N (%) | p-value |
|---|---|---|---|---|---|
| *Stigma* 0.508 | | | | | |
| Stigmatized | 125 (13.8) | 12 (3.2) | 92 (13.8) | 229 | |
| Not stigmatized | 8 (0.8) | 2 (0.54) | 7 (1.05) | 17 | |
| Missing | 767 (85.2) | 353 (96.1) | 564 (85.0) | 1684 | |
| *Quality of life: Median [Q1-Q3]* | | | | | |
| MCS | 50.2 [41.2-57.0] | 49.2 [40.8-55.3] | 50.9 [38.6-56.3] | 50.2 [49.7-50.5] | 0.403 |
| PCS | 53.1 [43.8-57.2] | 53.5 [44.8-57.3] | 52.7 [43.5-56.7] | 53.1 [52.9-53.3] | 0.548 |
| *Viral load* 0.020 | | | | | |
| ≤ 40 | 766 (85.0) | 336 (91.6) | 579 (87.3) | 1681 | |
| > 40 | 104 (11.5) | 26 (7.0) | 70 (10.5) | 200 | |
| Missing | 30 (0.5) | 5 (1.4) | 14 (2.1) | 49 | |

[a]Men who report having sex with other men.

[b]Immigrant > 10 years.

[c]Immigrant < 10 years.

[d]Alcohol Use Disorder Identification Test (AUDIT-10).

## Sample size calculations

As of December 31, 2022, 2155 individuals completed the OCS questionnaire in the ten different sites of the OCS. The primary outcome is adherence to ART, with 692 participants with suboptimal adherence and 1293 participants with optimal adherence. We planned a study with 1930 subjects. The sample size resulted in 80% power to detect a difference of 20% or greater between participants with suboptimal and optimal [35]. The Type I error probability associated with the test of the null hypothesis is 0.05 for two-tailed chi-squared statistic. (PS: Power and Sample Size Calculation version 3.1.2, 2014 by W.D. Dupont & W.D. Plummer Jr).

## Data analysis

Statistical analysis was performed using R software version 4.4.1. We used descriptive statistics to analyze participants' characteristics, reporting proportions for categorical variables and median with interquartile range (IQR) for continuous variables. The latter were compared using the Wilcoxon rank-sum test, as these scores are not normally distributed [36]. Chi-square tests were used for categorical variables.

 The variable selection was guided by a priori knowledge of their association with adherence to ART, considering collinearity between variables and potential confounding. Age, sex, ethnicity, employment, education level, substance use, stigma and living alone were included in all models as potential confounders, regardless of their significance [3,12,37,38]. The remaining variables were selected using stepwise model selection based on the Wald statistic from multiply imputed data [39,40].

## Regression analysis

We conducted a logistic regression analysis for dichotomous outcomes (adherence to ART and suppression of the viral load) and a multiple linear regression for continuous outcomes (quality of life). To check the model fit for multiple linear regression models, we used a pooled $R^2$. The models developed for logistic regression using the imputed dataset will not provide an Akaike Information Criteria (AIC) [40]. As we have taken a combine approach of using a prior defined variable, and Wald method, we believe the models are fit.

The dichotomous outcomes are reported as odds ratio (OR) and 95% confidence intervals, and continuous outcomes are reported as mean differences with 95% confidence intervals. The statistically significant level is set when the *p*-value is < 0.05 or the 95% CI excludes the null value.

The OCS questionnaire had data missing at random (MAR), so we performed ten imputations for each model and combined the results using Rubin's rules [39,40].

## Subgroup analysis

We performed subgroup analysis to assess the differences in health outcomes (adherence to ART, quality of life and viral load) of people living with HIV from different socio-demographics in Ontario, Canada. The data from the OCS Questionnaire 2022 was collected during the COVID-19 pandemic when virtual visits were first introduced. [15]. Since government COVID-19 policies and users' preferences may have impacted decisions to attend virtual visits, we compared outcomes during and after lockdowns [41].

## Results

In 2022, the OCS questionnaire was completed by 2155 people, with 1930 providing details on the type of care they received (S1 Fig). That year, 1021/1930 (53%) HIV care visits were conducted by telephone, 23/1930 (1.2%) via computer visits, and 1563/1930 (80%) were provided through in-person visits. The median age of the participants was 55 years [IQR: 45–62]. In 2022, stratum-specific proportions of participants in the three different types of care modalities (i.e., in-person, virtual and participants who used both in-person and virtual care) varied by participant characteristics (Table 1). Notably,1493/1930 (78%) participants were men, with the majority being MSM, compromising 58.6% (1131/1930). Regarding care modality preferences, in-person visits were most preferred across all genders. Table 1 provides the statistical relationship between all other variables and the type of care.

## Adherence to ART

There were 600 (66.6%) participants with optimal ART adherence. According to our logistic regression analysis, the odds of adherence to ART were higher for participants who used virtual care than in-person care in both adjusted and unadjusted analysis (OR 1.47%, 95% CI: 1.14, 1.89; AOR 1.31%, 95% CI: 1.00, 1.71).

## Viral load

There were 766 (85.0%) participants with adequate viral load suppression. The odds of viral load suppression were higher in the virtual group than in person-care (OR 1.81, 95%CI: 0.69, 2.85; AOR 1.67, 95% CI:1.03, 2.63).

## Quality of life

The average MCS score was 50.2, and the average PCS score was 53.1. According to our adjusted multiple regression model, combined virtual and in-person care is associated with an improved quality of life in terms of MCS compared to in-person care (Mean difference (MD) -0.90, 95% CI: 2.10, 0.30; adjusted MD 0.960, 95% CI: 0.05, 1.86). All results can be found in Table 2.

## Sub-group analysis

Among patients whose physicians based their preference of the type of care mode on the viral load, receiving in-person care was associated with lower odds of adherence to ART than receiving virtual care (OR 0.29, 95% CI: 0.10, 0.83; AOR 0.28, 95% CI: 0.09, 0.86).

Among patients whose physicians based their preference of the type of care mode on the viral load, the receipt of both virtual and in-person care was associated with decreased MCS quality of life compared with participants who used the

**Table 2. Multivariable regression analysis of HIV-related outcomes of people receiving care in Ontario, Canada.**

| Variable | Unadjusted Effect Estimates (95%CI) | p-value | Adjusted Effect Estimates (95% CI) | p-value |
|---|---|---|---|---|
| *Adherence to ART (OR)* | | | | |
| In-person(ref) | 1.00 (ref) | | 1.00 (ref) | |
| Virtual and in-person | 1.23 (0.99, 1.52) | 0.05 | 1.03 (0.82, 1.30) | 0.743 |
| Virtual | 1.47 (1.14, 1.89) * | 0.003 | 1.30 (1.00, 1.70) * | 0.048 |
| *Viral load (OR) <40 ml/cc preferred* | | | | |
| In-person(ref) | 1.00 (ref) | | 1.00 (ref) | |
| Virtual and in-person care | 1.13 (0.52, 1.56) | 0.431 | 1.08 (0.76, 1.53) | 0.651 |
| Virtual care | 1.81 (0.69, 2.85) * | 0.010 | 1.67 (1.03, 2.63) * | 0.0374 |
| *[1]MCS Quality of life* | | | | |
| In-person visit | 0 (ref) | | 0 (ref) | |
| Virtual and in-person | -0.90 (-2.10, 0.30) | 0.142 | 0.96 (0.05,1.86) * | 0.038 |
| Virtual | -0.53 (-1.99, 0.92) | 0.469 | 0.13 (-0.94, 1.23) | 0.810 |
| *[2]PCS Quality of life* | | | | |
| In-person visit | 0 (ref) | | 0 (ref) | |
| Virtual and in-person | -0.90 (-2.10, 0.30) | 0.142 | -0.91 (-1.24,0.85) | 0.711 |
| Virtual | -0.53 (-1.99, 0.92) | 0.469 | 0.09 (-1.14,1.32) | 0.811 |

[a]Adjusted for covariates.

*Statistically significant.

[1]Mental Component Summary Score.

[2]Physical Component Summary Score.

in-person care mode. (MD -5.60, 95% CI: -9.46, 1.75; Adjusted MD -3.75, 95% CI: -6.51, -0.99). Detailed findings are provided in S 1 Table.

## Discussion

This study examined three types of care used in the OCS cohort 2022: in-person, virtual and patients using both. Patients across all demographic and medical backgrounds used virtual care options. In this cohort, the participants who used virtual care mode most frequently were 51–60, perhaps suggesting that older patients are comfortable with technology. However, the predominant mode of care within this cohort remained in-person visits. In this cohort consisting of participants living with HIV in Ontario, we observed that participants with better adherence to ART and with viral load suppression preferred the virtual care, and participants with a higher MCS quality of life utilized both virtual and in-person care.

Sex was the strongest predictor of the type of care used, with male MSM favouring in-person care. Participants residing in Toronto preferred virtual care compared to those in Eastern Ontario, though these groups overlap, as many male MSM in Toronto might be the same individuals. Participants with any form of depression tended to use both virtual and in-person care, possibly choosing their care mode based on the severity of their depression at the time of the visit.

Our subgroup analysis based on physicians' preference for the type of visit by viral load revealed decreased ART adherence within the virtual care group. However, the primary analysis showed a 30% improvement in ART adherence among participants using virtual visits compared to in-person groups. This suggests physicians preferred in-person visits for patients with unsuppressed viral lows and poor ART adherence.

This study has several limitations that prevent drawing definitive conclusions. The cross-sectional design prevents us from establishing causality, making it difficult to determine whether virtual care directly influences HIV related-health outcomes. Additionally, the data analyzed is from 2022, a period when virtual care was newly introduced and still in the early stages of implementation, including the development of specific standards, equipment selection and user training [11],[41],

Moreover, during the study period, decisions regarding the type of care were often influenced by the government's lock-down orders rather than the preferences of physicians or patients [39]. Non-response bias further restricts the conclusion of differences between care modes. Furthermore, challenges related to optimizing virtual care usability and user-friendliness remain unexplored.

The OCS employs a standardized questionnaire developed in consultation with the OHTN, CAB and the governance committees. However, the primary outcome—ART adherence—is based on self-reporting, which is subjective and prone to recall bias [3,27].

The study's findings are influenced by selection and information biases, as participants are predominantly aged 51–55, with high incomes and stable housing. This limits the understanding of age-related acceptance and barriers. The cohort mainly includes highly engaged individuals, which doesn't fully represent overall care engagement. Younger, tech-savvy individuals and rural populations are underrepresented in this cohort, limiting insights into the broader applicability of virtual care and rural-urban differences [38].

Despite potential sample biases, the study's large sample size and real-time data collection enhance the robustness of its findings.

The study was guided by a CAB, whose input was integrated throughout the study, including their feedback on result interpretation. Collaborating closely with experienced HIV care physicians provides valuable insights into clinical practice dynamics and potential implementation strategies based on study outcomes [24].

### Future research

It's essential to recognize that this research is exploratory, and the COVID-19 pandemic lockdowns confound it. Future research should explore the impact of virtual visits on HIV care and improve access. Key issues include standards, licensure, equity, and payment systems. Virtual visits can help address physician shortages, especially in rural areas, by offering flexible, quality care. Expanding virtual care to all socioeconomic groups is vital. Redesigning care based on necessity and feasibility could enhance access and outcomes [42,43].Future studies should assess system-wide effects, user satisfaction, cost-effectiveness, and physician views while linking OCS data with ICES data to analyze clinical outcomes and patient preferences.

### Conclusion

In our study in the OCS 2022 of people living with HIV, participants with better self-reported adherence to ART and a suppressed viral load preferred virtual care as compared to in-person care. Moreover, participants who used both virtual and in-person care reported higher mental health quality of life.

### Supporting information

**S 1 Fig.  Study flow diagram of participants who received HIV care in 2022.**
(TIF)

**S 1 Table.  Multivariable subgroup regression analysis of HIV-related outcomes of people receiving care in Ontario, Canada.**
(DOCX)

### Acknowledgments

The OHTN Cohort Study Team consists of Dr. Ann Burchell (Interim Principal Investigator; email: Ann.Burchell@unity-health.to), Unity Health and University of Toronto; Dr. Anita Benoit (Co-Investigator), University of Toronto; Dr. Lawrence Mbaugbaw (Co-Investigator), McMaster University; Dr. Sergio Rueda, CAMH and University of Toronto; Dr. Gordon

Arbess, Unity Health; Dr. Corinna Quan, Windsor Regional Hospital; Dr. Curtis Cooper, Ottawa General Hospital; Elizabeth Lavoie and Dr. Maheen Saeed, Byward Family Health Team; Dr. Mona Loutfy and Dr. David Knox, Maple Leaf Medical Clinic; Dr. Nisha Andany, Sunnybrook Health Sciences Centre; Dr. Sharon Walmsley, University Health Network; Dr. Michael Silverman, St. Joseph's Health Care; Tammy Bourque, Health Sciences North; Dr. Marek Smieja, Hamilton Health Sciences Centre; Wangari Tharao, Women's Health in Women's Hands Community Health Centre; Holly Gauvin, Elevate NWO; Dr. Jorge Martinez-Cajas, Kingston Hotel Dieu Hospital; and Dr. Jeffrey Craig, Lakeridge Positive Care Clinic.

We gratefully acknowledge all of the people living with HIV who volunteer to participate in the OHTN Cohort Study. We also acknowledge the work and support of OCS Governance Committee (Aaron Bowerman, Adrian Betts, Barry Adam, Cornel Gray, Dane Record, Jasmine Cotnam, Jason Brophy, Mary Ndung'u, Rodney Rousseau, Ruth Cameron, YY Chen) OCS Scientific Steering Committee (Anita Benoit, Ann Burchell, Barry Adam, Curtis Cooper, David Brennan, Kelly O'Brien, Lance Mcready, Lawrence Mbuagbaw, Mona Loutfy, Pierre Giguere, Sean Hillier, Sergio Rueda (Chair), and Trevor Hart) and Indigenous Data Governance Circle (Meghan Young, Randy Jackson, Trevor Stratton). The OHTN also acknowledges the work of past Governance Committee and Scientific Steering Committee members.

We thank all interviewers, data collectors, research associates, coordinators, nurses, and physicians who provide data collection support. The authors also wish to thank OCS staff for data management, IT support, and study coordination: Lucia Light, Mustafa Karacam, Nahid Qureshi, and Tsegaye Bekele. The Ontario Ministry of Health supports the OHTN Cohort Study.

We also acknowledge the Public Health Laboratories, Public Health Ontario, for supporting record linkage with the HIV viral load database.

The OHTN Cohort Study is supported by the Ontario Ministry of Health and Long-Term Care.

The opinions, results and conclusions are those of the authors and no endorsement by the Ontario HIV Treatment Network or Public Health Ontario is intended or should be inferred.

Realize provided support and guidance for CAB recruitment, consultation on community engagement and study design.

## Author contributions

**Conceptualization:** Nadia Rehman, Lawrence Mbuagbaw, Dominik Mertz, Aaron Jones.

**Formal analysis:** Nadia Rehman.

**Investigation:** Nadia Rehman, Lawrence Mbuagbaw, Giulia M. Muraca, Aaron Jones.

**Methodology:** Nadia Rehman, Lawrence Mbuagbaw, Dominik Mertz, Giulia M. Muraca, Aaron Jones.

**Project administration:** Nadia Rehman, Lawrence Mbuagbaw, Aaron Jones.

**Resources:** Lawrence Mbuagbaw, Aaron Jones.

**Software:** Nadia Rehman, Aaron Jones.

**Supervision:** Lawrence Mbuagbaw, Dominik Mertz, Aaron Jones.

**Validation:** Dominik Mertz, Aaron Jones.

**Visualization:** Nadia Rehman, Aaron Jones.

**Writing – original draft:** Nadia Rehman.

**Writing – review & editing:** Lawrence Mbuagbaw, Dominik Mertz, Giulia M. Muraca, Aaron Jones.

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
