## [Decision Letter · Decision Letter 0]

2 Feb 2025

PONE-D-24-55188Association between virtual visits and health outcomes of people living with HIV: A cross-sectional studyPLOS ONE

Dear Dr. Rehman,

Thank you for submitting your manuscript to PLOS ONE. After careful consideration, we feel that it has merit but does not fully meet PLOS ONE’s publication criteria as it currently stands. Therefore, we invite you to submit a revised version of the manuscript that addresses the points raised during the review process.

We look forward to receiving your revised manuscript.

Kind regards,

Jianhong Zhou

Staff Editor

PLOS ONE

Journal Requirements:

2. For studies involving third-party data, we encourage authors to share any data specific to their analyses that they can legally distribute. PLOS recognizes, however, that authors may be using third-party data they do not have the rights to share. When third-party data cannot be publicly shared, authors must provide all information necessary for interested researchers to apply to gain access to the data. (https://journals.plos.org/plosone/s/data-availability#loc-acceptable-data-access-restrictions) 

3. One of the noted authors is a group or consortium: Ontario HIV Treatment Network Cohort Study and Realize

In addition to naming the author group, please list the individual authors and affiliations within this group in the acknowledgments section of your manuscript. Please also indicate clearly a lead author for this group along with a contact email address.

5. We notice that your supplementary figure is uploaded with the file type 'Figure'. Please amend the file type to 'Supporting Information'. Please ensure that each Supporting Information file has a legend listed in the manuscript after the references list.

**Additional Editor Comments:**

The manuscript has been assessed by two reviewers, and their comments are appended below. Could you please carefully revise the manuscript to address all comments raised?

Reviewers' comments:

Reviewer's Responses to Questions

**Comments to the Author**

1. Is the manuscript technically sound, and do the data support the conclusions?

Reviewer #1: Yes

Reviewer #2: Yes

2. Has the statistical analysis been performed appropriately and rigorously? 

Reviewer #1: No

Reviewer #2: Yes

3. Have the authors made all data underlying the findings in their manuscript fully available?

Reviewer #1: Yes

Reviewer #2: Yes

4. Is the manuscript presented in an intelligible fashion and written in standard English?

Reviewer #1: Yes

Reviewer #2: Yes

5. Review Comments to the Author

Reviewer #1: Feedback and Comments to Association between virtual visits and health outcomes of people living with HIV: A cross-sectional study

The manuscript titled "Association between virtual visits and health outcomes of people living with HIV: A cross-sectional study" has been thoroughly reviewed. This is a topic that is both intriguing and promising in the field of HIV retention in care. The work has significant potential for contribution to the field.

With careful consideration, I believe the paper needs some revisions and explanations. To enhance the quality and clarity of your research, I suggest the following:

Introduction: NA

Methods:

1. This study focuses on the HIV care that the patients received. Please be more specific about how the authors determine the category of patients’ care. For example, was the care type extracted from the medical record system? If the patient transitioned from in-person to virtual care, will it considered as “combination” or what?

2. The authors defined the optimal ART adherence as participants who reported never missing a dose or missing a dose more than 3 months ago, while the patients who missed doses more recently were considered as suboptimal adherence. Is there any other study that used this method? Personally, I prefer to consider the number of doses they took and missed to determine the ART adherence instead of the time since they missed a dose.

3. In “Secondary outcomes”, the authors talked about the scales/tools they used to measure QOL, please add a few sentences to mention if the higher score means better or worse. This will help readers who are not familiar with the assessment tools.

4. The authors mentioned the smoking status on page 9, lines 159-161. But this variable doesn’t show in any table or content after.

5. In the part of “data analysis”, the authors mentioned “majority selection method”. I’m not familiar with this method and couldn’t find useful information based on the reference “Heymans MaE I. Applied missing data analysis with SPSS and (R) Studio. Heymans and Eekhout;2019.”. Please explain this method.

6. Another thing the authors should think about is they mentioned the scores were not normally distributed in data analysis part. However, they used linear regression for regression analysis. One of the assumptions behind the linear regression model is normality. I understand the QOL scores might be normal, but in this situation, Table 1 should do ANOVA for these variables to avoid misunderstanding. The authors should be careful about these details and use the correct statistical analysis model.

Results:

1. In Table 1, please add a column reporting all. Move the p-values one line up to align with the variable names.

2. In Table 1, there are a couple of variables with very small samples under some categories (Northern Ontario under region, not stigmatized under stigma, for example). I am a bit concerned about the sample size here. Is it possible to re-categorize the groups to avoid this situation?

3. In Table 1, Alcohol use disorder syndrome is under the “Mental health conditions”. I’m not sure if this is correct.

4. Please re-format the CI in the writing. Some were written as “1.14, 1.89”, some were “0.69-2.85”. I personally would use “95% CI: (A, B)”.

5. In Table 2, under “Viral load (OR)”, virtual care is significant in unadjusted model and adjusted model. No “*” is marked here.

6. I personally suggest breaking Table 2 into 2 parts: for categorical outcome and continuous outcome separately.

Discussion:

1. The authors could talk about more why the virtual visits patients are more likely to have better ART adherence and viral load suppression.

2. Since this study found the virtual visit patients are more likely to have better adherence and treatment outcome, the authors should discuss why more patients still prefer in-person visit. The obstacle of transitioning from in-person to virtual care, and the benefit of virtual care.

Reviewer #2: there are several areas that could be improved or clarified.

- The introduction is dense with information, making it somewhat challenging to follow. Breaking it into smaller, clearer segments could enhance readability.

- While the introduction outlines the context of HIV care in Ontario and the introduction of virtual care, a brief overview of the significance of HIV in public health could set the stage more effectively.

- The introduction mentions "virtual care" but does not define what constitutes it beyond technical examples. A more comprehensive definition might aid readers unfamiliar with the concept.

- The mention of the limited evidence regarding the effectiveness of virtual care feels slightly abrupt. It could benefit from a brief example or a reference to prior studies that highlight this issue.

- Some transitions between sentences could be smoother. For instance, the shift from discussing the adoption of virtual care to its limitations is somewhat sudden.

- The objectives are clearly stated, but reiterating why these specific outcomes (adherence, QOL, viral load) are important could strengthen the rationale for the study.

- In the method section please add sample size calculation formula and provide details of calculations. It is not known what outcome was used.

Please add goofness of fit criteria for the logistc regression and also provide a confusion matrix for class predictions.

6. PLOS authors have the option to publish the peer review history of their article (what does this mean? ). If published, this will include your full peer review and any attached files.

**Do you want your identity to be public for this peer review?** For information about this choice, including consent withdrawal, please see our Privacy Policy .

Reviewer #1: No

Reviewer #2: No

---

## [Author Response · Author response to Decision Letter 1]

11 Feb 2025

The Editor,

PLOS ONE Journal

Dear Editor,

Thank you for the opportunity to revise our manuscript no. PONE-D-24-55188R1: “Association between virtual visits and health outcomes of people living with HIV: A cross-sectional study”, for publication in PLOS ONE Journal.

We sincerely appreciate the reviewers’ time and dedicated effort in evaluating our manuscript and providing their constructive feedback. We have carefully reviewed their commentary and have implemented their suggestions. Below, we outline our revisions to their specific points.

Following the reviewers’ feedback, we believe these revisions have significantly strengthened our manuscript's quality and scientific value. Attached is our revised paper and a marked-up version highlighting the changes addressed. We are confident that these changes adhere to this journal’s expectations and academic reputation, and we remain open to further constructive comments.

Thank you again for your consideration,

Sincerely,

Nadia Rehman

Reviewers’ Comments Authors’ Response Page No.

General Comments

Thank you for bringing this to our attention. We have ensured the manuscript complies with PLOS ONE’s style requirements, including file naming.

2. For studies involving third-party data, we encourage authors to share any data specific to their analyses that they can legally distribute. PLOS recognizes, however, that authors may be using third-party data they do not have the rights to share. When third-party data cannot be publicly shared, authors must provide all information necessary for interested researchers to apply to gain access to the data. (https://journals.plos.org/plosone/s/data-availability#loc-acceptable-data-access-restrictions) For any third-party data that the authors cannot legally distribute, they should include the following information in their Data Availability Statement upon submission:

We have expanded on the data retrieval methods and provided detailed information on obtaining data from the third party. The revised text is as follows:

“The data is obtained from OHTN from the OCS study, which focuses on four research areas: (i) social and behavioral, (ii) clinical, (iii) HIV prevention, and (iv) health services. Due to ethical restrictions, the authors cannot share the dataset publicly. However, data requests can be made by submitting a Research Application Process (RAP) to OHTN. For inquiries, contact OCS at ocs@ohtn.on.ca.”

Page 17-18,

Lines 290-293

3. One of the noted authors is a group or consortium: Ontario HIV Treatment Network Cohort Study and Realize

In addition to naming the author group, please list the individual authors and affiliations within this group in the acknowledgments section of your manuscript. Please also indicate clearly a lead author for this group along with a contact email address. Thank you for requesting this detail. We have updated the contact information, author list, and affiliations for the Ontario HIV Treatment Network. Page 18-19,

Line 306-33

4. Your ethics statement should only appear in the Methods section of your manuscript. If your ethics statement is written in any section besides the Methods, please delete it from any other section. We have incorporated this feedback and addressed ethics in the Methods section. It now reads as:

“Ethics

All participants provide written informed consent, and the cohort design and consent forms are approved by the University of Toronto Research Ethics Boards (REBs) and REBs at each participating site [1]. Data was accessed on November 23, 2023. And the data was in de-identified form.”

Page 7, Lines 115-118

5. We notice that your supplementary figure is uploaded with the file type 'Figure'. Please amend the file type to 'Supporting Information'. Please ensure that each Supporting Information file has a legend listed in the manuscript after the references list.

We have corrected the file name and labeled it as "Supporting Information." The legend for the supplementary documents is now provided after the references list.

Page 23, Line 451-452

Reviewer #1

Methods

1. This study focuses on the HIV care that the patients received. Please be more specific about how the authors determine the category of patients’ care. For example, was the care type extracted from the medical record system? If the patient transitioned from in-person to virtual care, will it considered as “combination” or what? We have clarified the missing details on the virtual care definition to ensure clarity. The updated text is:

“We categorized the HIV care patients received into three mutually exclusive categories: i). In-person care in the clinic ii). Virtual care either by telephone or video call, iii). Received both in-person and virtual forms of care. We defined virtual care as visits with the HIV care physician by telephone or video call, as defined in the OCS Questionnaire 2022.”

Page 8, Lines 139-143

2. The authors defined the optimal ART adherence as participants who reported never missing a dose or missing a dose more than 3 months ago, while the patients who missed doses more recently were considered as suboptimal adherence. Is there any other study that used this method? Personally, I prefer to consider the number of doses they took and missed to determine the ART adherence instead of the time since they missed a dose. We have refined our dichotomization and provided additional details on our exposure. Since we are using OCS data, we cannot modify how the variable is recorded. However, we have justified the appropriateness of this measure. The revised text is as follows:

“Adherence to ART is measured by self-report (never skipped, within the past week, 1-2 weeks, 2-4 weeks ago, 1-3 months ago, more than three months ago), in the standardized OCS questionnaire. This questionnaire is part of the Adult AIDS Clinical Trials Group (ACTG) adherence assessment and has been validated in previous studies [2-4].Self-reported nonadherence has shown high specificity, indicating its clinical significance and the need for further discussions between providers and patients [2]. For this study, we dichotomized adherence into optimal adherence (≥95%) and suboptimal adherence (<95%). We considered optimal adherence as participants who reported never missing a dose or those who missed a dose more than three months ago. In contrast, suboptimal adherence included those who missed doses within the past week, 1–4 weeks ago, 1-3 months, or responded with "don't know." These classifications were established in consultation with HIV specialists from a dedicated HIV care facility.”

Page 7, Lines 126-135

3. In “Secondary outcomes”, the authors talked about the scales/tools they used to measure QOL, please add a few sentences to mention if the higher score means better or worse. This will help readers who are not familiar with the assessment tools. We appreciate the reviewers for highlighting this important detail. We have made the changes, and now it reads as:

“QOL was assessed using the Short Form 12-item Health Survey (version 2), which includes the Mental Component Summary Score (MCS) and the Physical Component Summary Score (PCS). Both scores are reported separately, with higher values indicating better QoL in their respective domains.” Page 8,

Line 138-141

4. The authors mentioned the smoking status on page 9, lines 159-161. But this variable doesn’t show in any table or content after. Thanks for bringing this to our attention. We have removed the smoking variable from this manuscript. As the sample size of smokers was not sufficient, we didn’t add it to our model.

Now it reads as:

“Clinical variables were extracted on the following health conditions: alcohol use disorder syndrome use was defined as on the Alcohol Use Disorder Identification Test (AUDIT-10) with harmful alcohol measured from 10- items (a score of ≥8 regardless of gender/sex) [5], depression measured by the Patient Health Questionnaire (PHQ) scale with nine items [6], and diagnosis of mental health comorbidities was based on self-report in the OCS question.”

Page 9, Line 159-163

5. In the part of “data analysis”, the authors mentioned “majority selection method”. I’m not familiar with this method and couldn’t find useful information based on the reference “Heymans MaE I. Applied missing data analysis with SPSS and (R) Studio. Heymans and Eekhout;2019.”. Please explain this method. We used different methods for variable selection in the imputed dataset:

1. Majority

2. Stack

3. Wald

Details on these methods can be found in Flexible Imputation of Missing Data by Stef van Buuren (Section 5.4: Stepwise Model Selection). Since all methods selected the same variables, we chose to report only one in our manuscript. Given that the majority method is less familiar, we have now reported the Wald method.

The revised text now reads:

"The remaining variables were selected using stepwise model selection based on the Wald statistic from multiply imputed data." Page 10, Line 184-185

6. Another thing the authors should think about is they mentioned the scores were not normally distributed in data analysis part. However, they used linear regression for regression analysis. One of the assumptions behind the linear regression model is normality. I understand the QOL scores might be normal, but in this situation, Table 1 should do ANOVA for these variables to avoid misunderstanding. The authors should be careful about these details and use the correct statistical analysis model. Thank you for your thorough review of our work. Before proceeding with model building, we tested the assumptions of multiple linear regression by generating diagnostic plots in R using a regression model based on prior covariates, outcomes, and exposures. The diagnostic plots included:

1. Residual plot

2. QQ plot

3. Scale-location plot

4. Residuals vs. leverage

We identified some outliers, with a few Cook's distance values reaching 1. However, we did not exclude these data points, as they represent plausible and valuable information, with some participants potentially having extreme values.

Multicollinearity was assessed using the variance inflation factor (VIF < 10) and tolerance (>0.1).

The data met all the assumptions of multiple linear regression:

1. Existence

2. Independence

3. Homoscedasticity

4. Normality

5. Linearity

6. No multicollinearity

Due to slight skewness in the QoL data, we used the Wilcoxon Rank Sum Test, which is appropriate for comparing two groups with skewed data. Page 10, Line 177-178

Results

1. In Table 1, please add a column reporting all. Move the p-values one line up to align with the variable names. I agree with the reviewers that the information in the table needs to be better represented. We have moved the p-values to a new line to align with the variable names and added an additional column for the total. Table 1, Pages 11-13

2. In Table 1, there are a couple of variables with very small samples under some categories (Northern Ontario under region, not stigmatized under stigma, for example). I am a bit concerned about the sample size here. Is it possible to re-categorize the groups to avoid this situation? We used the OCS questionnaire 2022 for this analysis. The original questionnaire includes ten regional categories, which we have reclassified into four distinct regions of Ontario. This categorization is based on the cross-sectional nature of the study and provides readers with insights into the prevalence and use of care. Further breakdown would risk losing valuable information.

Regarding stigma, there were significant missing values. Given this, we only included stigma in the subgroup analysis. However, it was important to report the actual numbers: out of the participants who provided stigma data, 229 had stigma, which represents a considerable proportion (229/1930) of events. Table 1

Pages 11-13

3. In Table 1, Alcohol use disorder syndrome is under the “Mental health conditions”. I’m not sure if this is correct. Alcohol use disorder is a separate variable, and we have now aligned it with the other variables. Table 1, Pages 11-13

4. Please re-format the CI in the writing. Some were written as “1.14, 1.89”, some were “0.69-2.85”. I personally would use “95% CI: (A, B)”. Thank you for pointing out this inconsistency. We have updated all the 95% CIs to the format (A, B).

5. In Table 2, under “Viral load (OR)”, virtual care is significant in unadjusted model and adjusted model. No “*” is marked here. We have written the unadjusted viral load as 1.81 (0.69, 2.85)*. Table 2,

Page 15

6. I personally suggest breaking Table 2 into 2 parts: for categorical outcome and continuous outcome separately. Thank you for your feedback. Since there are only three outcomes, and we have clearly separated each outcome into distinct rows, we have kept it as is. Table 2, Line 14

Discussion

1. The authors could talk about more why the virtual visits patients are more likely to have better ART adherence and viral load suppression. We agree with this comment; however, due to the cross-sectional nature of the study, we were cautious about making conclusive statements regarding the use of any type of care. This study focuses solely on the association between the type of care and HIV-related health outcomes. Page 15-16, Lines

2. Since this study found the virtual visit patients are more likely to have better adherence and treatment outcome, the authors should discuss why more patients still prefer in-person visit. The obstacle of transitioning from in-person to virtual care, and the benefit of virtual care. This feedback comes from discussions with OCS experts and HIV care physicians, who advised that the study's findings should be non-conclusive and non-directive, given the cross-sectional nature of the study.

Reviewer # 2

Introduction

The introduction is dense with information, making it somewhat challenging to follow.

Breaking it into smaller, clearer segments could enhance readability. We have incorporated this change into the manuscript's introduction, removing redundant information and summarizing it.

Page 5-6, lines 58-88

While the introduction outlines the context of HIV care in Ontario and the introduction of virtual care, a brief overview of the significance of HIV in public health could set the stage more effectively. We understand this feedback. Our focus has been solely on the data from the OCS study, which includes its own participants. There may be differences in data from the public health sector, and supporting public health in this context is challenging, especially since in 2022, there was limited data available on virtual care elsewhere.

The introduction mentions "virtual care" but does not define what constitutes it beyond technical examples. A more comprehensive definition might aid readers unfamiliar with the concept. We have provided details on how we defined virtual care. The text now reads as:

” We categorized the HIV care patients received into three mutually exclusive categories: i). In-person care at the clinic ii). Virtual care either by telephone or video call, iii). Combination of in-person and virtual forms of care. We defined virtual care as visits with the HIV care physician by telephone or video call, as defined in the OCS Questionnaire 2022.”

Page 8, Lines 144-147

The mention of the limited evidence regarding the effectiveness of virtual care feels slightly abrupt. It could benefit from a brief example or a reference to prior studies that highlight this issue. We understand this feedback and have added more references to summarize the information. The text now reads as:

“Evidence on the effect of virtual care on health outcomes for PLHIV is limited. While some studies focus

---

## [Decision Letter · Decision Letter 1]

17 Apr 2025

PONE-D-24-55188R1Association between virtual visits and health outcomes of people living with HIV: A cross-sectional studyPLOS ONE

Dear Dr. Rehman,

Thank you for submitting your manuscript to PLOS ONE. After careful consideration, we feel that it has merit but does not fully meet PLOS ONE’s publication criteria as it currently stands. Therefore, we invite you to submit a revised version of the manuscript that addresses the points raised during the review process.

The reviewer has made additional comments. Therefore, I recommend minor revisions to address the comments.

We look forward to receiving your revised manuscript.

Kind regards,

Mohammad Mofatteh, PhD, MPH, MSc, PGCert, BSc (Hons), MB BCh (c)

Academic Editor

PLOS ONE

Journal Requirements:

**Additional Editor Comments:**

The reviewer has made additional comments. Therefore, I recommend minor revisions to address the comments.

Reviewers' comments:

Reviewer's Responses to Questions

**Comments to the Author**

1. If the authors have adequately addressed your comments raised in a previous round of review and you feel that this manuscript is now acceptable for publication, you may indicate that here to bypass the “Comments to the Author” section, enter your conflict of interest statement in the “Confidential to Editor” section, and submit your "Accept" recommendation.

Reviewer #1: All comments have been addressed

2. Is the manuscript technically sound, and do the data support the conclusions?

Reviewer #1: Yes

3. Has the statistical analysis been performed appropriately and rigorously? 

Reviewer #1: Yes

4. Have the authors made all data underlying the findings in their manuscript fully available?

Reviewer #1: Yes

5. Is the manuscript presented in an intelligible fashion and written in standard English?

Reviewer #1: Yes

6. Review Comments to the Author

Reviewer #1: I'm glad to see the comments for the first version are all be addressed/answered.

Here are some minor issues need editing:

1. Introduction: page 5 line 78, abbreviation PLHIV shows for the first time, needs the full term here.

2. The consistency of abbreviation: page 8, in the part of secondary outcomes, “QOL” and “QoL” are both used.

3. The consistency of number of digits in results and tables: page 11, some percentages keep 1 digit on the decimal and some are round numbers. Table one and two have the same issue.

4. Page 15, table 2, the variable “virtual care” is significant in adjusted model, needs a “*” after the CI.

7. PLOS authors have the option to publish the peer review history of their article (what does this mean? ). If published, this will include your full peer review and any attached files.

**Do you want your identity to be public for this peer review?** For information about this choice, including consent withdrawal, please see our Privacy Policy .

Reviewer #1: No

---

## [Author Response · Author response to Decision Letter 2]

17 Apr 2025

The Editor,

PLOS ONE Journal

Dear Editor,

Thank you for the opportunity to revise our manuscript (PONE-D-24-55188R1), titled “Association between virtual visits and health outcomes of people living with HIV: A cross-sectional study,” for consideration in PLOS ONE.

We sincerely appreciate the reviewers’ time and thoughtful feedback. We have carefully addressed each comment and incorporated the suggested changes to enhance the clarity, quality, and scientific rigor of our manuscript.

Below, we provide a detailed response to each point raised. The revised manuscript and a marked-up version highlighting the changes are attached. We believe these revisions align with the journal’s standards and welcome any further feedback.

Thank you again for your consideration.

Sincerely,

Nadia Rehman

Please find below the response to the reviewer’s comments.

Introduction

Abbreviation PLHIV shows for the first time, needs the full term here. We have made this correction by introducing the full term followed by the abbreviation. The revised text now reads:

“Evidence on the effect of virtual care on health outcomes for people living with HIV (PLHIV) is limited. While some studies focus on retention improvement rather than health outcomes, findings are mixed.” Page 5, line 78

The consistency of abbreviation: page 8, in the part of secondary outcomes, “QOL” and “QoL” are both used.

Thank you for pinpointing this mistake. We have corrected it by introducing the abbreviation for quality of life as QoL. We have fixed the error through out the manuscript. The revised text now reads:

“QoL was assessed using the Short Form 12-item Health Survey (version 2), which includes the Mental Component Summary Score (MCS) and the Physical Component Summary Score (PCS). Both scores are reported separately, with higher values indicating better QoL in their respective domains.” Page 8, line 139

Page 3, line 35,

Page 6, lines 87 & 91

The consistency of a number of digits in results and tables: page 11, some percentages keep 1 digit on the decimal, and some are rounded numbers. Tables one and two have the same issue.

We appreciate the reviewer’s careful attention to detail. We have made the necessary corrections in both tables, and all percentages are now presented with one decimal place. Page 11

The variable “virtual care” is significant in adjusted model, needs a “*” after the CI. We have made the correction and added an asterisk to virtual care in Table 2. Page 15, table 2

---

## [Editor Report · Decision Letter 2]

22 Apr 2025

Association between virtual visits and health outcomes of people living with HIV: A cross-sectional study

PONE-D-24-55188R2

Dear Dr. Rehman,

We’re pleased to inform you that your manuscript has been judged scientifically suitable for publication and will be formally accepted for publication once it meets all outstanding technical requirements.

Kind regards,

Mohammad Mofatteh, PhD, MPH, MSc, PGCert, BSc (Hons), MB BCh (c)

Academic Editor

PLOS ONE

Additional Editor Comments (optional):

The authors have responded well to the reviewers' comments. The manuscript has been improved and can be accepted.
---

## [Editor Report · Acceptance letter]

PONE-D-24-55188R2

PLOS ONE

Dear Dr. Rehman,

I'm pleased to inform you that your manuscript has been deemed suitable for publication in PLOS ONE. Congratulations! Your manuscript is now being handed over to our production team.

Kind regards,

on behalf of

Dr. Mohammad Mofatteh

Academic Editor

PLOS ONE